# Immobilized Bisphosphonates as Potential Inhibitors of Bioprosthetic Calcification: Effects on Various Xenogeneic Cardiovascular Tissues

**DOI:** 10.3390/biomedicines10010065

**Published:** 2021-12-29

**Authors:** Irina Y. Zhuravleva, Anna A. Dokuchaeva, Elena V. Karpova, Tatyana P. Timchenko, Anatoly T. Titov, Svetlana S. Shatskaya, Yuliya F. Polienko

**Affiliations:** 1E. Meshalkin National Medical Research Center of the RF Ministry of Health, 15 Rechkunovskaya Street, 630055 Novosibirsk, Russia; a_dokuchaeva@meshalkin.ru (A.A.D.); t_timchenko@meshalkin.ru (T.P.T.); 2N.N. Vorozhtsov Novosibirsk Institute of Organic Chemistry SB RAS, 9 Lavrentiev Avenue, 630090 Novosibirsk, Russia; karpovae@nioch.nsc.ru (E.V.K.); polienko@nioch.nsc.ru (Y.F.P.); 3V. Sobolev Institute of Geology and Mineralogy SB RAS, 3 Academician Koptyug Avenue, 630090 Novosibirsk, Russia; titov@igm.nsc.ru; 4Institute of Solid State Chemistry and Mechanochemistry SB RAS, 18 Kutateladze Street, 630128 Novosibirsk, Russia; shatskaya@solid.nsc.ru

**Keywords:** bioprosthetic calcification, bisphosphonate, porcine aortic wall, bovine jugular vein, cross-linking, glutaraldehyde, diepoxide

## Abstract

Calcification is the major factor limiting the clinical use of bioprostheses. It may be prevented by the immobilization of bisphosphonic compounds (BPs) on the biomaterial. In this study, we assessed the accumulation and structure of calcium phosphate deposits in collagen-rich bovine pericardium (Pe) and elastin-rich porcine aortic wall (Ao) and bovine jugular vein wall (Ve) cross-linked with glutaraldehyde (GA) or diepoxy compound (DE). These tissues were then modified with pamidronic (PAM) acid or 2-(2′-carboxyethylamino)ethylidene-1,1-bisphosphonic (CEABA) acid. Tissue transformations were studied using Fourier-transform infrared spectroscopy. After subcutaneous implantation of the biomaterials in 220 rats, calcification dynamics were examined using atomic absorption spectrophotometry, light microscopy after von Kossa staining, and scanning electron microscopy coupled with energy-dispersive X-ray spectroscopy The calcium content in all GA-cross-linked tissues and DE-cross-linked Ao increased to 100–160 mg/g on day 60 after implantation. BPs prevented the accumulation of phosphates on the surface of all materials and most effectively inhibited calcification in GA-cross-linked Ao and DE-cross-linked Pe. PAM containing -OH in the R1 group was more effective than CEABA containing -H in R1. The calcification-inhibitory effect of BPs may be realized through their ability to block nucleation and prevent the growth of hydroxyapatite crystals.

## 1. Introduction

Bioprosthetic heart valves (BHVs) have indisputable advantages over mechanical valves in terms of hemodynamics and the patient’s quality of life. Therefore, surgeons and patients are increasingly opting for bioprostheses. While in 2008, the share of mechanical prostheses was 45.3% of total aortic valve replacements, in 2017, it was only 17% [1]. Accordingly, the number of implanted BHVs increased nearly two-fold in the same period [2]. The main disadvantage of BHVs is the high rate of calcific dysfunction, which limits their use, especially in young patients. In the last decade, minimally invasive technologies, such as transcatheter heart valve replacement, have been actively developed. The number of transcatheter procedures is growing yearly and will continue to grow. Transcatheter heart valve leaflets are made from biological materials, mostly bovine or porcine pericardium; therefore, overcoming calcification is especially important for this progressive approach.

The conventional preservation method for cardiovascular tissue in commercial valves is to cross-link the collagen with glutaraldehyde (GA). However, it is well known that GA treatment leads to massive deposition of calcium phosphates in xenogeneic materials, and despite decades of active research, the mechanism of calcification remains unclear [3,4]. Methods to prevent bioprosthetic calcification, including the use of surfactants, trivalent cations, aminooleic acid, and alcohols, have not yet been introduced in surgery or have not received an unambiguous assessment of long-term clinical results [3,4,5,6]. For biomaterials consisting predominantly of collagen, the most effective approach is non-GA cross-linking using genipin, carbodiimide, epoxides, or methacrylates [7,8,9]. Clinical studies have shown that, even in infants, epoxy-treated bovine pericardium conduits are calcified substantially more slowly, although calcification cannot be completely avoided [10]. However, non-GA preservation is ineffective for elastin-rich materials, such as aortic and bovine jugular vein walls, which are used in stentless aortic bioprostheses and right-sided valved conduits [11,12]. Elastin does not react with cross-linkers such as GA and epoxides [13], suggesting that the calcium-binding sites in collagen and elastin are different. Indeed, in collagen, they appear only after exposure to GA, whereas in elastin, they are associated with natural regions of the molecule.

We believe that one of the more promising approaches to inhibit bioprosthetic calcification is the immobilization of bisphosphonates (BPs) on the biomaterial. The effectiveness of this approach was first demonstrated by Francis et al. [14] in 1969 and was experimentally substantiated in the 1980s [15,16]. It has been shown that the BP treatment of GA-preserved bioprosthetic materials can significantly reduce the calcification of xenogeneic valves in animal experiments [15,16,17]. However, the interest in immobilized BPs suddenly waned, maybe because pamidronic (PAM) and ethidronic acids were the only BP drugs available in those days. Moreover, the latter cannot be immobilized on biomaterials as they do not contain a reactive group.

At present, several BP drugs are available. All BPs contain a P-C-P moiety, which is a functional analogue of the P-O-P moiety of the natural calcification regulator, polyphosphate. Their mechanisms of action are diverse and depend on their molecular structure. However, they are all able to inhibit calcium phosphate nucleation (i.e., the formation of a calcium phosphate crystal nucleus) and hydroxyapatite (HAP) crystal growth owing to direct physicochemical interactions with calcium (Ca^2+^) and phosphates [18]. The central carbon of BPs is bound to two other groups, R1 and R2 (Figure 1A). R1 is generally a hydroxyl group, whereas the R2 structure varies widely. In particular, R2 may contain primary or secondary amines. This group can be used for BP immobilization by binding with an epoxy or aldehyde masking group that remains free after cross-linking of the biomaterial [15,16,19].

BPs are largely classified into two groups: nitrogen-containing BPs (N-BPs) and nitrogen-free BPs (non-N-BPs). The mechanism of action of systemically administered non-N-BPs is associated with the incorporation into metabolism of stable adenosine triphosphate analogs, leading to osteoclast apoptosis [20,21]. N-BPs have a more complex mechanism, involving farnesyl pyrophosphate, geranylgeranyl pyrophosphate, and G-proteins [22,23], and they affect not only the apoptosis of osteoclasts but also their differentiation and maturation [24].

The mechanism of action of immobilized BPs has not been studied. We hypothesized that it involves direct physicochemical effects in inhibiting calcium phosphate nucleation and crystallization. Using elastin-rich biomaterials implanted subcutaneously in rats, we previously demonstrated that the effectiveness of immobilized N-BPs depends on the composition of the extracellular matrix (ECM), the cross-linker used, and the chemical structure of the immobilized compound, rather than on the amount of immobilized BP [19]. In a screening of six N-BPs, we found that PAM and 2-(2′-carboxyethylamino)ethylidene-1,1-bisphosphonic acid (CEABA) (Figure 1) had the strongest inhibitory effect on the calcium-binding capacity of porcine aortic (Ao) and bovine jugular vein (Ve) walls, cross-linked with GA or ethylene glycol diglycidyl ether (DE) [19]. To unravel the mechanism of action of immobilized BPs, it is necessary to understand which changes they induce in the processes of accumulation and transformation of calcium phosphates in bioprosthetic materials. Therefore, in this study, we assessed the dynamics of accumulation and the structure of calcium phosphate deposits in collagen-rich bovine pericardium (Pe) and elastin-rich Ao and Ve xenogeneic materials cross-linked with GA or DE and modified with PAM or CEABA.

## 2. Materials and Methods

### 2.1. Chemicals

As cross-linking agents, we used 25% GA (catalog No. 253857, Panreac Química SLU, Barcelona, Spain) and 97% DE (N. Vorozhtsov Novosibirsk Institute of Organic Chemistry, SB RAS, Novosibirsk, Russia). PAM and CEABA were synthesized according to previously described procedures [19,25,26].

### 2.2. Biomaterial Treatment

#### 2.2.1. Biomaterial Pre-Treatment with Cross-Linking Agents

Fresh porcine aortic wall (Ao), bovine pericardium (Pe), and bovine jugular vein wall (Ve) were obtained from healthy animals immediately after slaughter. They were rinsed several times with 0.9% NaCl and preserved at room temperature in either 0.625% GA (0.1 M phosphate buffer, pH 7.4) for 21 days, with solution refreshments on days 2 and 7, or 5% DE (0.1 M phosphate buffer, pH 7.4) for 14 days, with a solution refreshment on day 3. 

#### 2.2.2. Development of the BP Treatment Method

To establish the time intervals, temperature, and solution concentration required to obtain optimal quantitative BP/tissue interaction, PAM and CEABA, and Ve were used ([19], Supplementary Information). 

In the first step, biomaterial samples preserved with DE were immersed in 0.05 M PAM solution (biomaterial/solution mass ratio, 1:15) at 25 °C (*n* = 60) or 37 °C (*n* = 60). After 4, 8, and 12 h, 20 samples were collected and rinsed with sterile 0.9% NaCl for 20 min. Ten samples were used for quantitative BP assessment, with the remaining 10 samples immersed in fresh 0.9% NaCl to remove all unbound drug for the assessment of PAM quantity. The sorption/desorption times were 4 h/4 h, 8 h/8 h, and 12 h/12 h. The results were plotted in a graph (Figure 2A).

In the second step, the optimal treatment concentration and duration for PAM and CEABA were determined. The optimal treatment time at 37 °C for PAM and CEABA was 5 h and 8 h, respectively, followed by washing off the unbound substances for 4 h. The concentrations of the solutions tested were 0.005, 0.05, 0.1, and 0.2 M. 

#### 2.2.3. Biomaterial Treatment with BPs

After preservation with GA or DE, all three biomaterials were thoroughly rinsed with sterile 0.9% NaCl three times for 20 min and then placed in a 0.05 M aqueous solution of PAM or CEABA in 0.1 M phosphate buffer (pH 7.3) at 37 °C for 8 h, followed by 4 h of washing. Then, all samples were stored in the alcohol solution (1% (octane-1,2-diol + 2-phenoxyethan-1-ol) and 20% ethanol) [27] until implantation.

### 2.3. Quantitative BP Assessment

The phosphorus content of the biomaterials preserved in the presence of GA or DE and treated with PAM or CEABA (*n* = 180) was evaluated before implantation using an atomic emission spectrometer (IRIS Advantage, Thermo Scientific “TJA Solution,” Waltham, MA, USA). The content of each BP was calculated as: [BP] (μM/g dry tissue) = ([P]m − [P]n)/62 × 1000, where [P]n is the phosphorus content (μg/mg) in the non-modified GA- or DE-cross-linked biomaterial and [P]m is the phosphorus content in the samples treated with a particular BP. 

### 2.4. Fourier-Transform Infrared (FTIR) Spectroscopy

Twenty-one biomaterial groups were studied: untreated Ao, Ve, and Pe, biomaterials preserved with GA or DE, and biomaterials further modified with PAM or CEABA. FTIR data were obtained using a Tensor 27 spectrometer (Bruker, Ettlingen, Germany). Biomaterial composition was identified based on absorption spectra (4000–400 cm^−1^) collected as KBr pellets by averaging 64 scans with a resolution of 4 cm^−1^. One milligram of sample was dispersed in 150 mg of KBr, with the mixture placed in a 13-mm pellet die and pressed. The attenuated total reflection (ATR-FTIR) spectra were obtained using a ZnSe crystal (angle of incidence, 45°). Each spectrum is the result of 100 co-added interferograms, in the range 4000–860 cm^−1^. Distilled deionized water was used to generate background spectra. Spectral manipulations were performed using the OPUS 6.0 software (Bruker). All spectra were normalized to the amide I band, assuming that the content of amide groups in proteins is the same before and after treatment. FTIR spectra of the BP solutions were recorded using ATR cells filled with 5 × 10^−2^ M solutions (pH 7, adjusted using sodium hydroxide solution). 

### 2.5. Subcutaneous Implantation of the Biomaterials in Rats

All experimental procedures were performed in accordance with EU Directive 2010/63/EU for animal experiments and were approved by the Ethics Committee of E. Meshalkin National Medical Research Center (protocol No. 05-2019, 15 March 2019). 

Eighteen groups of biomaterials were tested: Ao/Ve/Pe cross-linked with GA or DE as controls and GA- and DE-cross-linked Ao/Ve/Pe modified with PAM or CEABA. The implants were washed with 0.9% NaCl for 20 min before implantation.

Four-week-old male Wistar rats (70–80 g, *n* = 220) were anesthetized with sevoflurane. Six incisions were made on the dorsal surface to prepare subdermal pouches. Each pouch was filled with one biomaterial sample (6 mm × 6 mm) and closed with one stitch. On days 10, 20, and 30 after implantation, 10 samples of each group were explanted and rinsed with 0.9% NaCl. Six samples were fixed in 10% neutral-buffered formalin for histological analysis. Four samples were fixed in a straightened state and dried at room temperature under sterile conditions for scanning electron microscopy (SEM) coupled with elemental analysis (EDS). The remaining 60-day explants (*n* = 190) were dried at 60 °C for calcium analysis.

### 2.6. Calcification Analysis

The explanted 60-day biomaterial samples were dried, weighed, and hydrolyzed in 14 M HNO_3_. Calcium was quantified using a Thermo Solaar M6 atomic absorption spectrophotometer (Thermo Fisher Scientific, Waltham, MA, USA).

### 2.7. Histological Analysis

All explanted samples fixed in formalin were embedded in paraffin and cut into 6-μm-thick sections. To observe calcium phosphate, the sections were subjected to von Kossa staining, i.e., treatment with 5% AgNO_3_ for 30 min and then counterstained with eosin (pink staining). The sections were then examined under a light microscope (Axioskop 40, Carl Zeiss Microscopy, Göttingen, Germany).

### 2.8. SEM–EDS Analysis

All samples were covered with a conductive carbon layer of 10–15 nm thick using a Quorum Q150T ES device (Quorum Technologies, Laughton, UK). They were then examined using a high-resolution scanning electron microscope MIRA 3 LMU with a Schottky cathode (Tescan Orsay Holding, Brno, Czech Republic) coupled with an INCA Energy 450+ X-Max-80 energy-dispersive spectrometer (Oxford Instruments Nanoanalysis, Abingdon, Oxfordshire, UK). The following settings were selected: accelerating voltage of 20 kV, probe current of 1.5 nA, electron beam diameter of 8–9 nm, spectral acquisition time of 20 s. The size of the X-ray generation zone in calcifications was 2–3 μm. EDS was performed only in electron-dense areas suspected of containing calcium phosphates. The analysis was performed in the spot mode or by scanning areas of up to 10 × 10 μm^2^. Pure elements or simple synthetic and natural compounds were used as reference samples. Corrections for the matrix effect were calculated by the XPP method, using the manufacturer’s software. Based on the data obtained, the Ca and P amounts (wt%) and Ca/P atomic ratio in each group of samples were calculated.

### 2.9. Statistical Analysis

Quantitative data were processed using Dell Statistica 13.0 (Dell Software, Aliso Viejo, CA, USA). As the distributions for the majority of the groups were not normal, non-parametric statistics was used. Quantitative data are reported as medians and interquartile (25–75%) ranges. The Mann–Whitney U-test was used to compare two groups and the Kruskal–Wallis test was used to compare three or more groups. The level of significance was set to *p* < 0.05.

## 3. Results

### 3.1. PAM and CEABA Content in Modified Bioprosthetic Materials

The treatment technology analysis results for Ve demonstrated that tissue saturation with the BP drugs was faster at 37 °C. Moreover, a larger amount of PAM was sorbed at this temperature (Figure 2A) and the amount of bound drug that was not desorbed from the tissue during long desorption was higher. PAM binding increased with increasing temperature, indirectly indicating a covalent attachment of the amino BP to the tissue. We considered the optimal temperature for BP modification of the biological tissue to be 37 °C, since it yielded better irreversible binding of the amino BP to the tissue. A higher temperature is inappropriate as the treatment lasts several hours and there is a risk of thermal damage to the structural proteins of the tissue.

For both PAM and CEABA, an increase in the concentration from 0.05 to 0.2 M significantly (*p* < 0.01) increased the amount of drug bound after 5 h of treatment (Figure 2B,C). Increasing the treatment time to 8 h significantly (*p* < 0.05) improved the binding for both BPs. The binding of PAM and CEABA after 8 h of treatment was the most significantly (*p* < 0.001) improved when the concentration was increased from 0.005 M to 0.05 M. It should be noted that the quantitative binding of the drugs after 8 h of treatment was in a rather narrow range of 20–25 μM/g dry tissue. We believe that this is additional evidence of the limited drug binding with free reactive groups of the cross-linking agent. Based on the combined findings, we concluded that in order to achieve effective modification, it is necessary to treat biomaterials with a BP solution with a concentration of at least 0.05 M for 8 h at 37 °C.

Quantitative BP assessment demonstrated that 8 h sorption + 4 h desorption allowed immobilizing 5–25 μg BP per 1 g of biological tissue (Figure 2D). The lowest BP content was found in Pe samples, while the modification efficiency did not depend on the pre-treatment method nor on the structure of the BP. The largest immobilized BP content was found in Ve. In this tissue also, pre-treatment did not affect the efficiency of modification, although an insignificant (*p* > 0.05) increase in immobilized BP content was observed when using PAM. The most interesting features were observed for Ao: GA-cross-linked Ao (GA-Ao) adsorbed PAM most actively (*p* < 0.001), whereas DE-cross-linked Ao (DE-Ao) better absorbed CEABA (*p* < 0.038). In addition, DE-Ao contained a significantly (*p* < 0.001) greater amount of each BP.

### 3.2. FTIR Spectroscopy Results

In the range 1300–900 cm^–1^, BP anions were characterized by two intensive wide bands at 1158 and 1073 cm^–1^ in the spectrum of PAM and three intense bands at 1173, 1139, and 1062 cm^–1^ in the spectrum of CEABA (Table 1, Appendix A). Comparison of the ATR-FTIR spectra of cross-linked and BP-treated biomaterials and spectra of the BP solutions (Figure 3) revealed the appearance of bands in the region 1170–1130 cm^–1^ in the spectra of the biomaterials studied. The band at 1070–1060 cm^–1^ is associated with intrinsic protein absorption and the absorption of DE and GA moieties included in the cross-linked protein structure. 

In line with findings in our previous study [18], we observed a complex band in the difference FTIR spectra of BP-treated biomaterials versus those only cross-linked (Figure 4). 

The negative band at 1430 cm^–1^ in the negative region of the difference spectra of all treated Ao and Ve samples may be attributed to bending vibrations of OH groups. It may be associated with changes in the energy as a result of the formation of hydrogen bonds between peptide chain hydroxide groups and BPs. A second negative band at 1508–1503 cm^–1^ (NH_2_ deformation) in the same FTIR difference spectra indicated a decrease in amino groups in the BP-treated samples through conversion into -NH_2_^+^ cations (Figure 4). The difference spectra of BP-treated and cross-linked Pe showed somewhat different changes in the biomaterial structure. Bands in the negative region close to those found in the difference spectra of the modified Ve and Ao were observed only in the spectrum of the Pe-GA-PAM sample. In the difference spectra of the remaining samples, the negative bands were shifted (Table 2). Further, an additional negative band appeared at 1557 cm^–1^, which coincided with the amide II band. The large spectral differences can be explained by the different compositions of the biomaterials. Together, these results showed that a large amount of BPs bound to Ao and Ve biomaterials non-covalently, through van der Waals interactions.

### 3.3. Calcium Deposit Dynamics

According to histological analysis, the samples treated with GA were the most mineralized. Primary foci of calcification arose in the deep layers of all studied tissues but their localization varied: 60–150 μm subendothelially and 250–270 mm subadventitially in Ao; no deeper than 50–60 μm from the intimate surface of Ve; and in the very central part in Pe cross-sections. From these primary foci, calcification gradually spread outward in all directions. Even materials modified with CEABA demonstrated rapid calcification, with crystal structures appearing in all types of GA-cross-linked materials by day 10 after implantation. 

GA-Ao showed the highest level of mineralization (Figure 5). On day 10 after implantation, GA-Ao samples showed small crystals that were evenly dispersed throughout the tissue and associated with elastin and smooth muscle cells. By day 20, the calcium deposits reached the outer margins of some samples. By day 30, major structural changes occurred in GA-Ao: severe crystallization damaged the fiber architecture and overall sample structure, with multiple fiber ruptures and splits visible across the tissue section. BP treatment decreased the number of calcium deposits at all time points; a morphological picture of (GA + CEABA)-Ao 30 days after implantation was similar to that of GA-Ao 20 days after implantation. In (GA + CEABA)-Ao on day 30, the spreading of mineralization intensified, with most deposits being small, scattered, and irregularly shaped.

GA-Pe also mineralized strongly (Figure 6). As early as day 10 after implantation, we observed massive calcium phosphate deposits in collagen fibers that formed large clusters as of day 20. On day 30, GA-Pe demonstrated a high level of calcification that affected the morphological characteristics of the tissue, as observed for GA-Ao. (GA + CEABA)-Pe was saturated with scattered dust-like particles on day 20, with further progression of crystal growth until day 30. 

GA-Ve samples were also subject to intense calcification. Calcium phosphate crystals were detected as early as day 10 after implantation. By day 20, elastin fibers were covered with mineral clusters that disturbed fiber orientation and disrupted the tissue into striped fragments. On day 30, elastin fibers were ruptured and assimilated into large calcium deposits, and calcified collagen could be observed. Treatment with CEABA suppressed cluster development, resulting in less structural damage on days 10 and 20 after implantation, although the number of initial calcium deposits on day 10 was equal to that in non-modified GA-Ve. A few GA-CEABA samples retained their non-mineralized structure for up to 30 days (Figure 7).

DE-Ao also strongly mineralized but initial points of crystallization were observed only on day 20 after implantation (Figure 5). They strictly followed the elastin fibers in the tissue and on day 30, they formed large, solid, ribbon-like clusters that caused fiber disruption and splitting. In (DE + CEABA)-Ao, the first signs of calcification, i.e., isolated dust-like particles, were visible on day 20 after implantation. By day 30, they developed into scattered crystals that were evenly distributed over the section surface. 

Mineralization in (DE + CEABA) samples was less significant than that in (GA + CEABA) samples. DE-treated Ve or Pe did not demonstrate any significant histological evidence of mineralization in the 30-day experiment.

PAM treatment significantly reduced the mineralization rate in both the GA-Ao and DE-Ao groups. In (GA + PAM)-Ao, initial single calcium deposits were observed on day 30. (DE + PAM)-treated samples were more calcified than their (GA + PAM)-treated counterparts. Initial calcium deposits were observed in these samples on day 20, and by day 30, scattered calcium-phosphate crystals formed that did not cause significant damage to the overall sample structure. In (GA + PAM)-Pe and (GA + PAM)-Ve, no calcium deposits were observed during the entire follow-up period. Together, these findings demonstrated that BP modification impedes the process of mineralization in bioprosthetic materials, with PAM modification being the most effective. 

Calcium accumulation in GA-Ao, -Ve, and -Pe lasted up to 60 days (Figure 8A). A similar trend was observed for DE-Ao. In DE-Pe tissue, the Ca concentration was 46-fold lower than that in GA-Pe (*p* = 0.0000) and did not change for up to 4 months (Figure 8B). Thus, it can be hypothesized that Pe tissue reaches its maximum calcium-binding capacity in 2 months. The calcium concentration in DE-Ve was three-fold lower than that in GA-Ve (*p* = 0.0002) and did not differ significantly (*p* > 0.05) from that in (GA + BP)- and (DE + BP)-treated samples (Figure 8A). BP-Ao tissue showed the most unexpected results when considering the histological data on the 30-day calcification dynamics: BP modification was substantially more effective for GA-cross-linked tissues than for their DE-cross-linked counterparts. BP modification suppressed calcium accumulation by 30-fold in GA-Ao but by only 4–8-fold in DE-Ao, with the calcium concentration in the latter similar to that in (DE + BP)-Ve.

### 3.4. SEM and EDS Analysis Results

We identified general trends in the accumulation and transformation of calcium and phosphates on the surfaces of the samples (Figure 9). In GA-Ao, the calcium deposits observed on day 10 were composed of calcium-deficient HAP (Ca/P = 1.34). As the content of both Ca and P increased progressively, the Ca/P ratio increased only to 1.4 by day 30. This non-stoichiometric HAP phase having Ca^2+^ vacancies indicates the potential for further Ca accumulation. In GA-Pe, Ca and P dynamics were similar to those observed in GA-Ao: on day 10, the Ca phosphate on the surface of the samples consisted of HAP (Ca/P = 1.51), but a progressive increase in the P content and an increase in Ca/P up to 1.7 occurred between days 20 and 30. The Ca content in GA-Ve increased more slowly than that of their Ao and Pe counterparts up to day 20. However, in the following 10 days, it increased very rapidly and accordingly, on day 30, it exceeded that of GA-Ao and leveled that of GA-Pe. The P content increased faster than the Ca content, which led to an increase in the Ca/P ratio from 0.25 to 20.39 between days 10 and 20. However, by day 30, the Ca and P content, as well as the Ca/P ratio of 1.35, suggested the presence of unsaturated HAP. By day 60, the Ca content in GA-Ve was only 20 mg/g lower (*p* = 0.055) than that in GA-Ao (Figure 8).

Modification with BPs differentially affected the Ca and P interactions of the different biomaterials. On day 10, P prevailed (Ca/P = 0.2–0.4) in the particles on the surface of (GA + BP)-Ao, whereas Ca prevailed (Ca/P = 3.3–6.3) in (GA + BP)-Pe. The extremely low (<1 wt%) content of Ca and P in (GA + BP)-Ao samples led us to conclude that treatment with BPs prevents HAP formation. In (GA + BP)-Pe, the Ca content increased from 1.3–1.4 (day 10) to 6.8–8.0 wt% (day 30), with a stable P level of <1 wt%. Such hypercalcium particles are probably carbonates or carbonate apatites. The calcification-inhibitory effect lasted at least until day 60, when the calcium level was 2–3 mg/g dry tissue in (GA + BP)-Ao and 3–6 mg/g in (GA + BP)-Pe (Figure 8). Electron-dense particles on (GA + BP)-Ve surfaces after 10 days featured with prevailing Ca but by day 20, their P content increased. By day 30, CEABA- and PAM-treated samples demonstrated differences in both Ca and P content: particles on PAM-treated surfaces contained significantly less Ca (*p* = 0.0036) and P (*p* = 0.0002), and had a lower Ca/P ratio (0.51 versus 1.46) than CEABA-treated samples.

For DE- and (DE + BP)-Pe, the 30-day dynamics of the Ca-P interactions did not dramatically differ from those for (GA + BP)-Pe: P content, which fluctuated within <2 wt% throughout the experiment, whereas the Ca content increased to 3–8%. This led to an increase in Ca/P to 3.5–4.5 in all groups of DE-pretreated Pe, suggesting the absence of HAP in these samples. A generally similar dynamics was found for DE- and (DE + BP)-Ve samples when compared with GA + PAM samples. This finding was confirmed both histologically and by SEM–EDS. 

However, DE-Ao behaved entirely differently. In this tissue, calcification was inhibited up to 20 days, after which Ca started to intensely accumulate, prevailing over P. By day 60, DE-Ao did not significantly differ in Ca content (*p* = 0.71) from the most calcified tissue, GA-Pe. After 60 days, the efficiency of immobilized BPs in inhibiting DE-Ao calcification was significantly lower than that for GA-Ao. PAM was more effective (*p* = 0.002) than CEABA, although for 30 days, the histological image and Ca concentration in surface particles were nearly the same for PAM- and CEABA-modified Ao.

It should be noted that the EDS results were calculated based on the analysis of surface electron-dense particles (Figure 10 and Appendix A). Therefore, these results are not indicative of the quantitative accumulation of Ca in the implant tissue but allow us to evaluate the interactions of Ca and P ions upon contact with the various biomaterials studied.

Very rarely, Al-Si or Fe-Ni particles (Figure 10A, spectrum 1) were observed. The former were considered dust particles that had fallen on the material surface during sample preparation, while the latter were considered to result from manipulations with surgical forceps when removing the implants and cleaning off the surrounding tissue. However, at all points in time and on all samples, we observed electron-dense particles containing a large amount of Ca in the absence of P or with a small amount of P (Figure 10A, spectrum 2; Figure 10D, spectrum 1). We regarded these as calcium carbonates, since there were no other elements in the spectra hinting at insoluble compounds with Ca. However, we cannot assert this with certainty because, firstly, carbon was sprayed onto the material surfaces and was abundantly represented in all spectra and, secondly, the organic substrates themselves (biological tissues) are rich in carbon. On day 10, such particles were especially abundant on GA- and (DE + BP)-Ao, as well as on GA- and (GA + BP)-Pe (Appendix A). On day 20, the amount of Ca particles decreased in these samples with a simultaneous increase in Ca phosphate deposits, but increased in (GA + BP)-Ao and GA-Ve (Appendix A). On day 30, significant surface areas of DE-Ao (Figure 10D) and GA-Pe (Appendix A) were occupied by Ca phosphates, namely HAP of varying degrees of saturation, and did not contain particles with a predominant Ca content. However, we observed numerous hypercalcium deposits on the surfaces of (GA + BP)-Ao and (GA + BP)-Pe, as well as DE- and (DE + BP)-Pe.

## 4. Discussion

To our knowledge, this is the first study to investigate how immobilized BPs affect the interaction of Ca and P ions with bioprosthetic tissues and to shed light on the mechanisms underlying their calcification-inhibitory activity. These results confirmed our previous findings that the intensity and dynamics of bioprosthetic calcification depend on the composition of the biomaterial ECM, the nature of the cross-linking agent, and the structure of the BP used for modification [12]. 

The quantitative accumulation of calcium in the biomaterials was investigated by the atomic absorption spectrometry on day 60 because, during this period, bioprosthetic materials accumulate calcium to amounts comparable to those observed in BHVs in patients [28]. In histological analysis, calcium phosphate accumulation was assessed for 30 days, as visual assessment allows us to evaluate not only the quantitative dynamics but also the calcium affinity of each tissue component. We took into account that the initial deposits spread centrifugally and gradually increased in mass, until they surfaced. The surface of most samples remained free of calcium conglomerates for up to 30 days, which made it possible to assess the features of each biomaterial in interaction with Ca and P ions, which penetrate the surface into the deep layers from the interstitial fluid of the recipient. 

We are the first to study these interactions by SEM and EDS of surface electron-dense particles to evaluate their Ca and P content. The interpretation of the results is rather complicated as both the quantitative distribution of particles on the surface and the mass percentages of Ca and P and atomic Ca/P ratio must be considered. Only by considering all these indicators, together with histological data and the levels of calcification after 60 days, can we evaluate the processes of ion–tissue interactions and the effect of cross-linking agents or BPs on them. The data obtained in this study allowed us to trace two major trends that require an interpretation of their genesis. 

(1) The routine treatment method for bioprosthetic materials is GA cross-linking; therefore, GA-treated materials were included as controls. Numerous electron-dense particles were distributed on the surfaces of GA-Ao and GA-Pe as early as day 10 after implantation, and despite the low Ca and P weight percentages, the Ca/P atomic ratios were typical for HAP, indicating that nucleation of this mineral was actively taking place. The amount of calcium phosphates increased over 30 days, as was clearly observed histologically. Further increases in the Ca and P content in electron-dense surface particles, as well as in the degree of saturation of HAP (Ca/P), occurred simultaneously. The calcification process in GA-Ve was different. On day 10, phosphates predominated in the surface particles, whereas after this time point, calcium preferentially accumulated, and by day 30, most of the surface was covered with unsaturated HAP (Ca/P = 1.35), indicating a potential for not only increasing crystal mass but also further Ca absorption. 

BPs prevented the accumulation of phosphate on the surface (and most likely in the deep layers) of all tissue. DE cross-linking had a similar effect. We believe that the effects of immobilized BPs are associated with their direct physicochemical effects on Ca and P ions and depend on the chemical structure of the BPs. According to classical concepts [18], the first and main phase of calcification is nucleation, in which Ca and P combine into a strong ionic complex, forming a crystal core. The second phase involves crystal growth. This occurs in GA-bioprosthetic tissues. BPs can prevent the deposition of phosphates from biological fluids containing calcium and phosphates in a metastable state (in the subcutaneous rat model, this is interstitial fluid; in human patients, it is blood). This effect may be the result of the negative gradient in the concentration of phosphate groups, an excess of which prevents the penetration of “external” phosphates into the surface and deep layers of implants, which complicates HAP nucleation. The decrease in the phosphate content in electron-dense particles on BP-modified surfaces indirectly confirms this hypothesis. A second possible mechanism is the strong binding of the HAP nucleus by phosphonate groups, which prevent crystal growth [18,29]. In this case, the R1 structure plays an important role because -OH enhances the binding of BP to Ca, in contrast to -H [29,30]. This is confirmed by our results: PAM with -OH in R1 was more effective than CEABA having -H in R1, as judged from the histological image on day 30. Both mechanisms may work simultaneously. It is unlikely that the PAM amino group contributes to the calcification-inhibitory effect [30], since it is spent on binding to the biological tissue during immobilization [18]. However, in CEABA, the secondary amine in the center of the aliphatic chain is such a reactive group (Figure 1) and the terminal carboxyl can take part in Ca binding.

(2) Modification with BPs was effective for both GA-Pe and DE-Pe for 60 days, as well as for GA-Ao. The calcification of DE-Ve, as well as (GA + BP)-Ve and (DE + BP)-Ve, was very similar (*p* > 0.05) and may be associated with a greater accumulation of phosphates in Ve than in other materials. Our results for Ao contradict those obtained in a previous study [31]. These researchers showed that pre-treatment with an epoxy compound (triglycidylamine) coupled with BP modification (2-(2-mercaptoethylamino)ethylidene-1,1-bisphosphonic acid) [32] significantly inhibited Ao calcification when compared with GA-pretreated Ao treated with the same BP. In our study, modification with PAM and CEABA was significantly (*p* < 0.05) more effective for GA-Ao than for epoxy-pretreated Ao. Further, it is difficult to unambiguously interpret the results obtained for Ve, where it was possible to achieve a decrease in calcification only three times, and equally for (GA + BPs)-, DE-, and (DE + BPs)-samples. 

We hypothesize that the effectiveness of BPs largely depends on the type of their chemical bonding with biological tissue. In turn, this mechanism depends on the structure of the tissue; that is, on the type of protein pattern of ECM. Pe has a homogeneous ECM that is composed exclusively of collagen and, therefore, the results for this tissue are predictable, reproducible, and fairly uniform. We have previously proved that PAM and CEABA can covalently bind to the reactive groups of GA and DE [19]. In this case, BPs can enhance the effect of epoxy cross-linking or block the adverse effects of GA. However, there are not many masking groups in the intermolecular and interfibrillar cross-linking of collagen. In our study, Pe binds a fairly low amount of BPs (especially, DE-Pe), mainly via covalent bonds with cross-linker masking groups. FTIR experiments (Figure 4) showed that the interaction of BPs with amino groups of the substrates is not typical for Pe, with the exception of Pe-GA-PAM. At the same time, these interactions are obvious for all types of aortic and venous tissue, the ECM of which contains not only collagen but also a lot of elastin, which is difficult to cross-link [13,33]. The presence of non-collagen proteins and their steric transformations during cross-linking [34] are responsible for decreasing the activity of BPs, despite the higher content of bound BPs in these elastin-rich tissues (Figure 2D). The collagen/elastin ratio varies widely depending on the individual, species, nutrients, and other characteristics. In addition, there are numerous other proteins associated with elastin fibers, such as fibrillin, vitronectin, fibulin, and laminin [35]. Any of these proteins can contain additional binding sites for BPs through the formation of H-bonds. It is known that PAM can thus interact with its amino group with threonine [23]. However, based on our FTIR results, which showed a decrease in amino groups in elastin-containing substrates, the most likely is the formation of H-bonds between the phosphate groups of BPs and amino groups of substrates. These can be the protein backbone amino groups and those of the Arg sidechain, with the involvement of serine oxygen in the formation of H-bonds [36,37]. This explains to some extent why the greater binding of BPs by the aortic and vein walls is accompanied by a lower anti-Ca effect as compared to Pe.

### Limitations

The present study was a first attempt to shed light on the mechanisms underlying the calcification-inhibitory effects of immobilized BPs. The main limitation of this study is that we analyzed whole tissues, which in addition to the main ECM proteins (collagen and elastin), contain numerous other components such as proteins, glycoproteins, and glycosaminoglycans. All of these can, in one way or another, affect the adsorption and binding of BP compounds. Therefore, we determined the mechanisms of their binding and action only hypothetically. To accurately unravel these mechanisms, model studies with purified collagen and elastin, and possibly also polylysine, are needed, which is the subject of further research.

## 5. Conclusions

BPs can be immobilized on bioprosthetic materials through covalent and non-covalent bonds. The calcification-inhibitory effect of immobilized BPs is realized through their physicochemical ability to block nucleation and prevent the growth of HAP crystals. The efficiency of immobilized BPs depends on the composition of the ECM of the bioprosthetic material, and the cross-linker used for its pre-treatment; BPs are most effective for GA-Ao and DE-Pe. PAM containing -OH in R1 is more effective than CEABA harboring -H in R1.

## Figures and Tables

**Figure 1 biomedicines-10-00065-f001:**
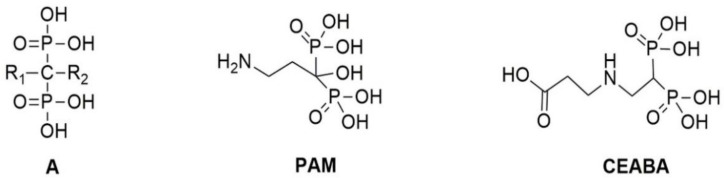
General structure of bisphosphonates (BPs) (**A**), pamidronic acid (**PAM**), and 2-(2′-carboxy-ethylamino)ethylidene-1,1-bisphosphonic acid (**CEABA**).

**Figure 2 biomedicines-10-00065-f002:**
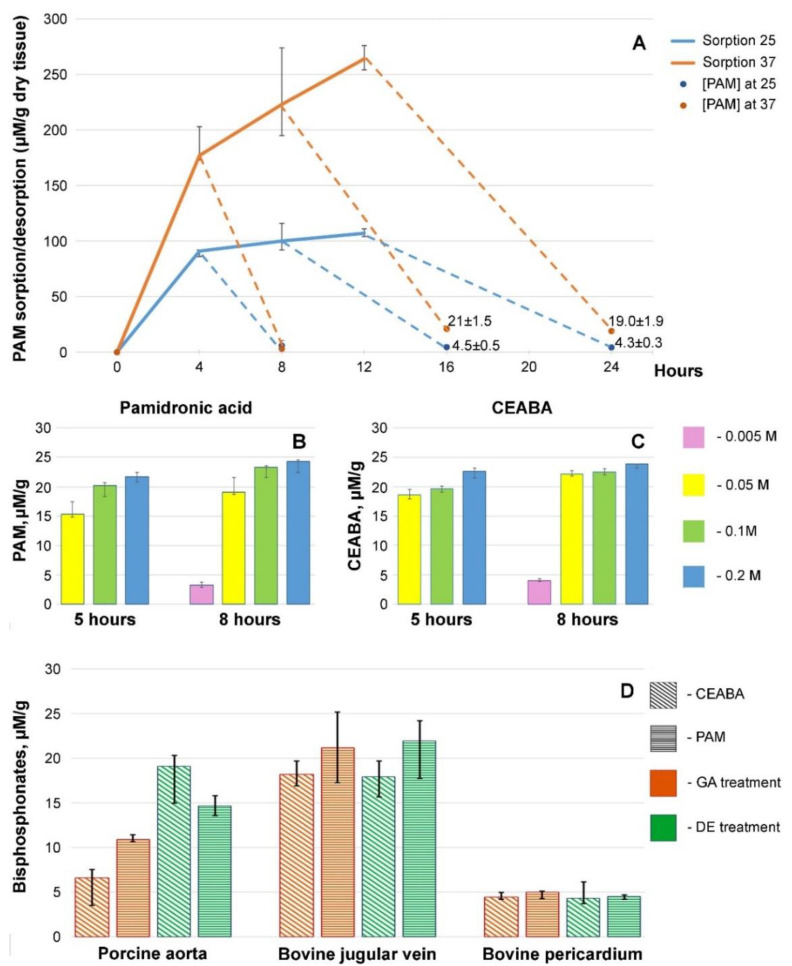
Dynamics of PAM sorption by DE-Ve and desorption of unbound drug. (**A**) Sorption at 25 °C (blue line) and desorption at 25 °C (blue dotted line); sorption at 37 °C (orange line) and desorption at 37 °C (orange dotted line). Amounts of bound PAM (**B**) and CEABA (**C**) (µM/g dry tissue) in bovine jugular vein wall after 5 h and 8 h treatment with the BPs at different concentrations at 37 °C. (**D**) BP concentrations in the biomaterials studied.

**Figure 3 biomedicines-10-00065-f003:**
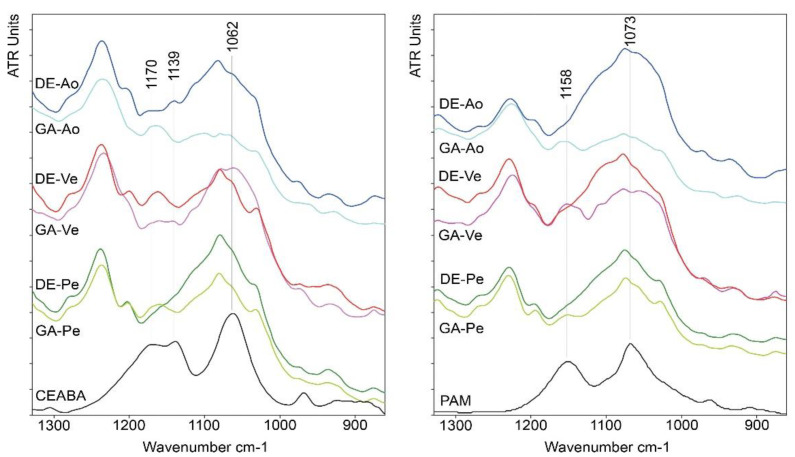
ATR-FTIR spectra of the BP solutions at pH 7 and of the cross-linked biomaterials treated with PAM and CEABA in the region 1340–860 cm^⁻1^.

**Figure 4 biomedicines-10-00065-f004:**
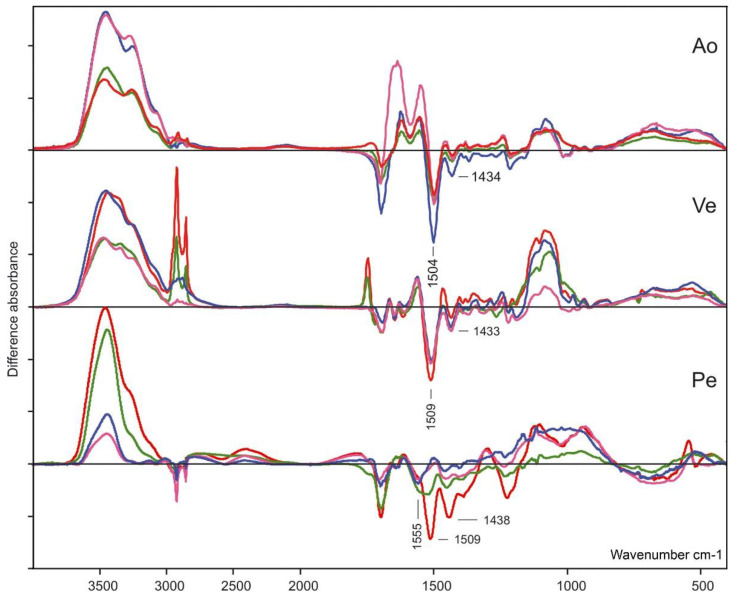
Difference FTIR spectra of the cross-linked biomaterials treated with PAM and CEABA (GA + PAM—red line, GA + CEABA—green line, DE + PAM—blue line, DE + CEABA—pink line).

**Figure 5 biomedicines-10-00065-f005:**
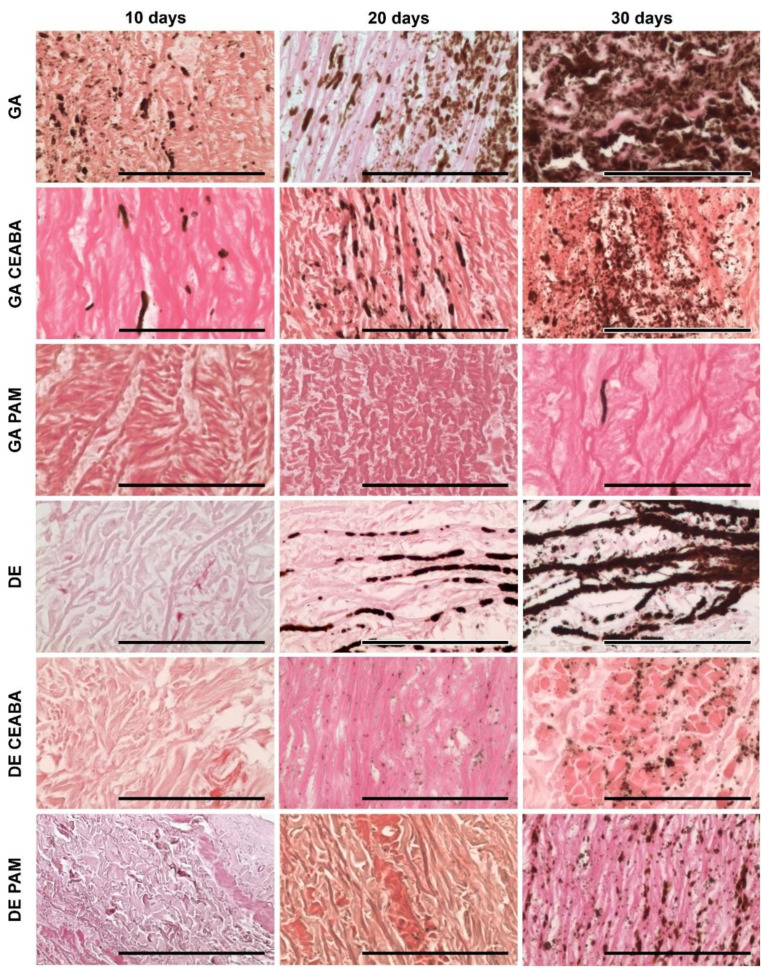
Calcium phosphate deposits in porcine aortic wall implanted subcutaneously in rats as visualized by von Kossa staining. Calcium phosphates are stained from light to dark brown and were localized predominantly along the elastin fibers or in smooth muscle cells. Scale bars: 100 μm.

**Figure 6 biomedicines-10-00065-f006:**
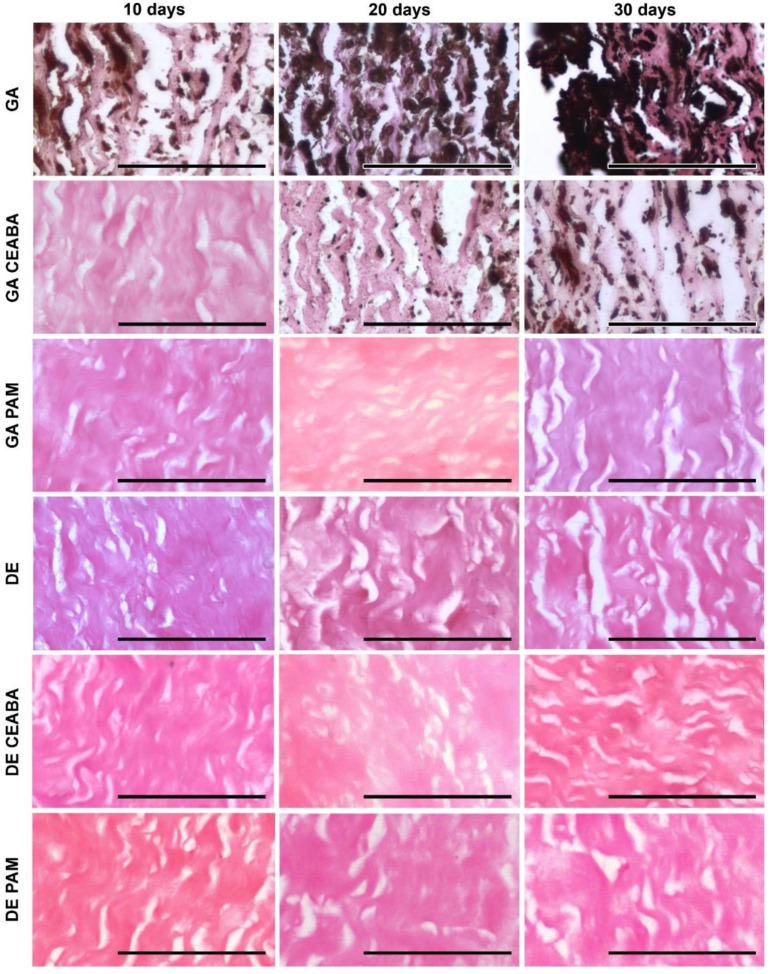
Calcium phosphate deposits in bovine pericardium implanted subcutaneously in rats, as visualized by von Kossa staining. Calcium phosphates were associated with collagen fibers in GA- and (GA + CEABA)-treated samples. All DE- and (GA + PAM)-treated samples were free of calcific deposits up to 30 days after implantation. Scale bars: 100 μm.

**Figure 7 biomedicines-10-00065-f007:**
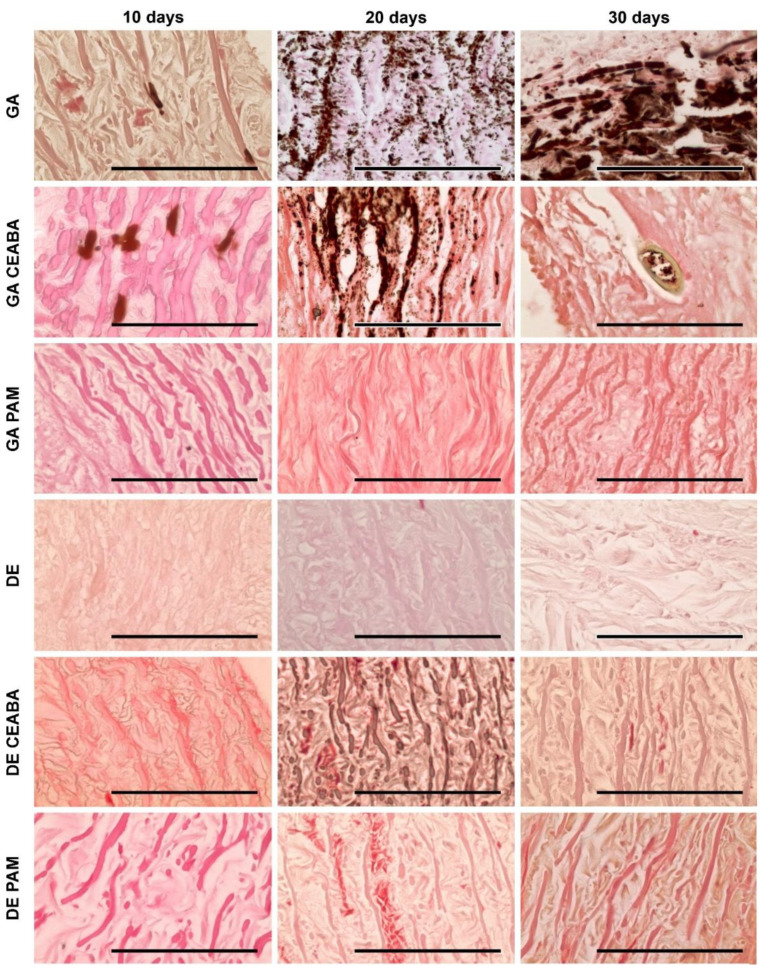
Calcium phosphate deposits in bovine jugular vein wall implanted subcutaneously in rats, as visualized by von Kossa staining. Calcium phosphates were observed in GA- and (GA + CEABA)-treated samples. Deposits are stained from light to dark brown and were localized predominantly along the elastin fibers or as dust-like particles in collagen. Large conglomerates of calcium phosphates were observed in GA-treated samples on day 30 after implantation. All DE- and (GA + PAM)-treated samples were free of calcific deposits. Scale bars: 100 μm.

**Figure 8 biomedicines-10-00065-f008:**
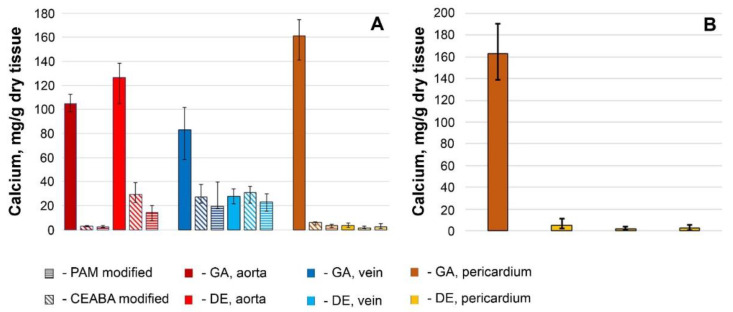
(**A**) Calcium content in all studied biomaterials on day 60 after implantation. (**B**) Calcium content in bovine pericardium on day 120 after implantation.

**Figure 9 biomedicines-10-00065-f009:**
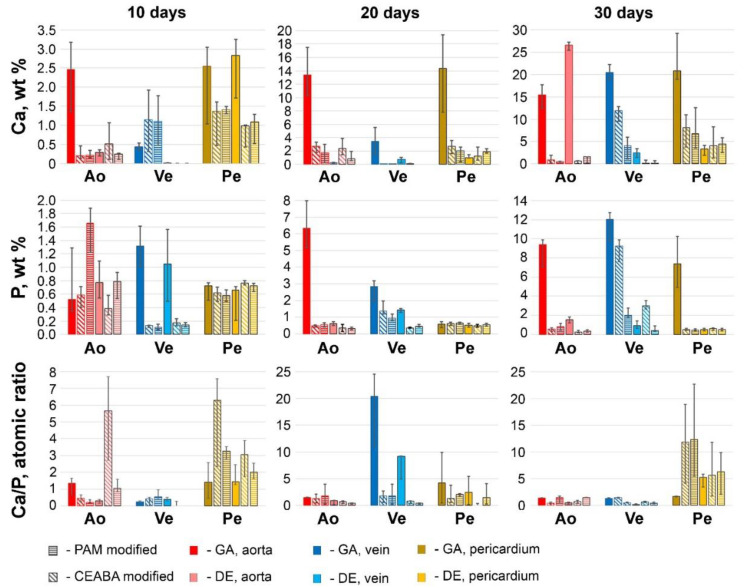
Calcium (Ca) and phosphorus (P) levels (wt%) and Ca/P atomic ratio in electron-dense particles on the implant surfaces.

**Figure 10 biomedicines-10-00065-f010:**
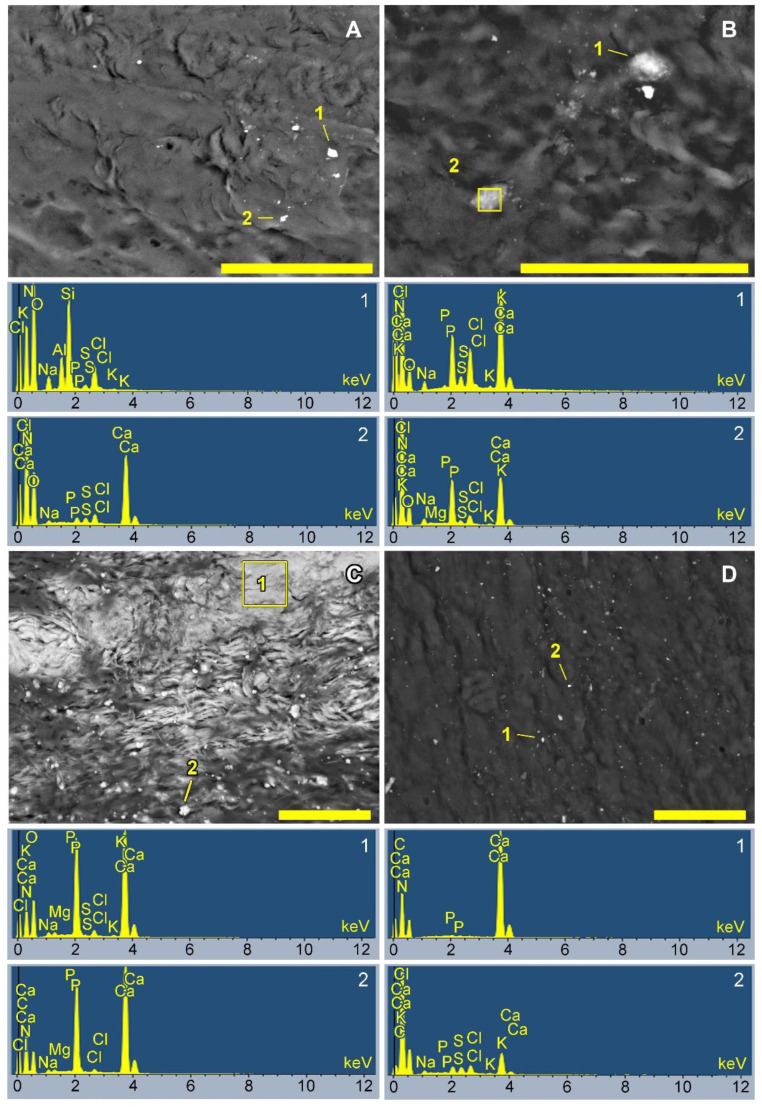
Surface SEM images of GA-Ve 10 days after implantation (**A**), DE-Ve 30 days after implantation (**B**), GA-Ao 30 days after implantation (**C**), and (GA + PAM)-Pe 30 days after implantation (**D**). Each image is accompanied by two characteristic elemental spectra of spots or areas indicated on the corresponding image. Scale bars: 50 μm.

**Table 1 biomedicines-10-00065-t001:** The assignments of ATR-FTIR peaks for PAM and CEABA in aqueous solution at pH = 7.

Wavenumber (cm^−1^)	PAM Assignment	CEABA Assignment
3000–2800	ν C–H	ν C–H
1573	-	ν_as_ COO^-^
1513	δ NH2	-
1468	δ CH2	δ CH2
1405	-	ν_s_ COO-
1172	-	ν P=O
1158	ν P=O +ν_as_ P–O of HPO_3_^−^	-
1139	-	ν_as_ P–O of HPO_3_^−^
1073	ν_as_ P–O of PO_3_^2−^	-
1062	-	ν_as_ P–O of PO_3_^2−^
968	ν_s_ P–O in PO_3_^2−^	-

**Table 2 biomedicines-10-00065-t002:** Displacement of negative bands in the difference spectra of the treated pericardium.

Sample	OH Deformation	NH_2_ Deformation
Pe-GA-PAM	1439	1508
Pe-GA-CEABA	1447	1517
Pe-DE-CEABA	1452	1529
Pe-DE-PAM	1454	1529

## Data Availability

The data presented in this study are available within the article.

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
