# Peer review of "Immobilized Bisphosphonates as Potential Inhibitors of Bioprosthetic Calcification: Effects on Various Xenogeneic Cardiovascular Tissues"

_biomedicines, 2021, doi:10.3390/biomedicines10010065_

Round 1

Reviewer 1 Report

This work investigated the effect of immobilized BPs on the interaction of Ca and P ions with bioprosthetic tissues, trying to uncover the inhibition mechanism of calcification. Various methods were employed to verify their hypothesis. The experiments are well designed. The manuscript is well orgnized with in-depth discussions. The reviewer believes this paper can be published in biomedicines with minor revisions.

1) Figure 4 is too large. The authors are suggested to revise.

2) The authors are suggested to specify the covalent bonding through adding more discussions.

Author Response

This work investigated the effect of immobilized BPs on the interaction of Ca and P ions with bioprosthetic tissues, trying to uncover the inhibition mechanism of calcification. Various methods were employed to verify their hypothesis. The experiments are well designed. The manuscript is well orgnized with in-depth discussions. The reviewer believes this paper can be published in biomedicines with minor revisions.

Dear Reviewer,

Thank you for substantial but benevolent comments. We tried to make the necessary changes.

  • Figure 4 is too large. The authors are suggested to revise.

Thank you. We have reduced Fig. 4.

2) The authors are suggested to specify the covalent bonding through adding more discussions.

Thank you. We have slightly changed and supplemented the text of the Discussion. It seems now more reasonable and understandable.

Reviewer 2 Report

This manuscript studied the accumulation and structure of calcium phosphate deposits in collagen-rich bovine pericardium and elastin-rich porcine aortic wall (Ao) and bovine jugular vein wall. The results indicated that the BPs can be immobilized on bioprosthetic materials through covalent and non-covalent bonds. The calcification-inhibitory effect of immobilized BPs is realized through their physicochemical ability to block nucleation and prevent the growth of HAP crystals.  This manuscript reports certain academic value and innovation.  However, there are some issues of the manuscript need to be attention. Authors may see the comments as the follow:

(1) In introduction section, it is better to provide more information about immobilization of bisphosphonates. Why to choose them? What is the advantage of these materials?

(2) It needs to simplify and condense some sentences. Such as “assessed the accumulation and structure of calcium phosphate deposits in collagen-rich bovine ericardium (Pe) ……1-bisphosphonic (CEABA) acid”.

(3) The evaluation model is different from the actual application scenario. How to interpret this result? “Subcutaneous implantation of the biomaterials in rats”.

(4) In Figures 5, 6, 7, and 10, the Scale bars are too big in the images. The authors can display its in 50 um.

(5) It is suggested to move the section 6. Limitations before the conclusions.

(6) Some recently research about the bioprosthetic or implants are suggested to be cited in this study, because they are focused on biocompatibility or applications in clinical, it shows potential inspiration on biomaterials interface design.

  • Xuan Pei, Lina Wu, et.al. 3D printed titanium scaffolds with homogeneous diamond-like structures mimicking that of the osteocyte microenvironment and its bone regeneration study. Biofabrication 2021,13: 015008.
  • Boqing Zhang, Huan Sun, et.al. 3D printing of calcium phosphate bioceramic with tailored biodegradation rate for skull bone tissue reconstruction. Bio-Design and Manufacturing. 2019, 2: 161–171.
  • Lina Wu, Changchun Zhou, et.al. Construction of biomimetic natural wood hierarchical porous structures bioceramic with micro/nano whisker coating to modulate cellular behavior and osteoinductive activity. ACS Appl. Mater. Interfaces 2020, 12, 48395-48407.

Author Response

Dear Reviewer,

Thank you for substantial but benevolent comments. We tried to answer your questions and made the necessary changes.

  • In introduction section, it is better to provide more information about immobilization of bisphosphonates. Why to choose them? What is the advantage of these materials?

 We have added a short explanation to this section (see Lane 70-72)

  • It needs to simplify and condense some sentences. Such as “assessed the accumulation and structure of calcium phosphate deposits in collagen-rich bovine ericardium (Pe) ……1-bisphosphonic (CEABA) acid”.

Thank you. We have done it.

  • The evaluation model is different from the actual application scenario. How to interpret this result? “Subcutaneous implantation of the biomaterials in rats”.

Subcutaneous implantation in rats is a conventional model developed in the 1980s for the primary screening of anti-calcification strategies [1-5]. In this model, “implanted valves and cusps were found to undergo mineralization showing histopathological features comparable to those of failed valve implants in humans, but with more accelerated kinetics (3 to 6 months for circulatory models and about 8 weeks for subcutaneous models)” [6, see section In vivo and in vitro experimental calcification of aortic valves (p.2131)]. All researchers agree that the absence of calcification obtained on this model is not a guarantee of the absence of cardiovascular bioprosthetic calcification in patients; however, the presence of calcification in rats indicates a high calcium-binding potential of the tested material and makes it possible to exclude it from further developments.

  1. Levy RJ, Wolfrum J, Schoen FJ, et al. Inhibition of calcification of bioprosthetic heart valves by local controlled-release diphosphonate. Science 1985;228:190-2;
  2. Jones M, Eidbo EE, Hilbert SL, et al. Anticalcification treatments of bioprosthetic heart valves: in vivo studies in sheep. J Card Surg 1989;4:69-73;
  3. Schoen FJ, Levy RJ, Nelson AC, et al. Onset and progression of experimental bioprosthetic heart valve calcification. Lab Invest 1985;52:523-32.
  4. Schoen FJ, Hirsch D, Bianco RW, et al. Onset and progression of calcification in porcine aortic bioprosthetic valves implanted as orthotopic mitral valve replacements in juvenile sheep. J Thorac Cardiovasc Surg 1994;108:880-7.
  5. Fishbein MC, Levy RJ, Ferrans VJ, et al. Calcifications of cardiac valve bioprostheses. Biochemical, histologic, and ultrastructural observations in a subcutaneous implantation model system. J Thorac Cardiovasc Surg 1982;83:602-9.
  6. A. Bonetti, M. Marchini, F. Ortolani. Ectopic mineralization in heart valves: new insights from in vivo and in vitro procalcific models and promising perspectives on noncalcifiable bioengineered valves. J Thorac Dis 2019;11(5):2126-2143

  • In Figures 5, 6, 7, and 10, the Scale bars are too big in the images. The authors can display its in 50 um.

Thank you. We have done it in Figs.5-7. In Fig.10, the images are presented with different magnification, which was done to facilitate the perception of information. It will be more difficult for readers to analyze information in the aggregate of images, checking the scale each time.

  • It is suggested to move the section 6. Limitations before the conclusions.

Thank you. We have done it.

  • Some recently research about the bioprosthetic or implants are suggested to be cited in this study, because they are focused on biocompatibility or applications in clinical, it shows potential inspiration on biomaterials interface design.

Xuan Pei, Lina Wu, et.al. 3D printed titanium scaffolds with homogeneous diamond-like structures mimicking that of the osteocyte microenvironment and its bone regeneration study. Biofabrication 2021,13: 015008.

Boqing Zhang, Huan Sun, et.al. 3D printing of calcium phosphate bioceramic with tailored biodegradation rate for skull bone tissue reconstruction. Bio-Design and Manufacturing. 2019, 2: 161–171.

Lina Wu, Changchun Zhou, et.al. Construction of biomimetic natural wood hierarchical porous structures bioceramic with micro/nano whisker coating to modulate cellular behavior and osteoinductive activity. ACS Appl. Mater. Interfaces 2020, 12, 48395-48407.

 We have read these articles with great interest and highly appreciated them. However, it should be noted that all of them are devoted to cell interactions with polymer or metal bone implants. Accordingly, the methods used in these researches for surface modification, aim to INCREASE ossification and attract calcium phosphates to the target zone. Our task is exactly the opposite, we must PREVENT the deposition of calcium phosphate on the tissue structures of animal body materials. That is, having different tasks, different substrates and different approaches to modification, we substantiate the ideology of our work in different ways. Thus, we could not find in what context these works can be cited. At the same time, we draw your attention to the fact that almost a half of the works cited in the "Introduction" were published in the last 3 years, that is, they are also new, but they relate specifically to prostheses from biological origin tissues.

Reviewer 3 Report

In their manuscript entitled "Immobilized bisphosphonates as potential inhibitors of bioprosthetic calcification: effects on various xenogeneic cardiovascular tissues" the authors desribe a an ver interesting problem in medicine an biomaterials development.
The mansucript is very well written and is very informative; results are very well presentend and supportetd by graphs and images.
As an suggestion for improving; the others coud think about to visialize by a sketch - or chemica structure formula, how the Caclium ist a) bound to the phospate (sure vio the Phospahte group) but suggest a modell how this is connectied to the tissue.
However, I recommend this paper for publihing; the authors deminstrated a very good work.

Author Response

Dear Reviewer,

We thank you for the high opinion of our work.

We will definitely use your suggestion to visualize our hypothesis about the calcium / phosphate interaction. Now we are preparing the next article, in which this hypothesis has a stronger support by a lot of results and experiments.

Round 2

Reviewer 2 Report

This manuscript is a resubmission of an earlier submission. The following is a list of the peer review reports and author responses from that submission.